# GRAPH FOUNDATION MODELS: BRIDGING LANGUAGE MODEL PARADIGMS AND GRAPH OPTIMIZATION

## ABSTRACT

The pretrain-transfer paradigm, which underpins the success of large language models (LLMs), has demonstrated the immense power of creating foundation models that learn generalizable representations from vast datasets. However, extending this paradigm to Operations Research (OR) problems on graph structures remains challenging due to the fundamental conflict between the statistical flexibility of language and the strict combinatorial constraints of graphs. To bridge this gap, we introduce the Graph Foundation Model (GFM), the first framework capable of solving all distance-based optimization problems on graph structures. By introducing the LLM-like self-supervised pre-training paradigm on the paths generated from random walks in the graph, GFM is compelled to internalize the graph's complex topological and combinatorial rules, where the connectivity of the structure itself can be treated as the supervisory signal. Unlike existing neural methods that learn complex and task-specific solving policies, our approach leverages the pre-trained GFM as a foundational model of the graph's intrinsic structure, which in turn enables a simple generative heuristic to tackle a diverse range of optimization challenges effectively. Comprehensive experiments on networks ranging from 20 to 893 nodes demonstrate that GFM achieves competitive performance against specialized solvers across a variety of distinct optimization task classes, while maintaining significantly faster inference times. Our work establishes a new paradigm of adapting the pretrain-transfer framework to graph optimization, opening the door for applying foundation model innovations to operations research.

## 1 INTRODUCTION

The remarkable success of large language models (LLMs) has cemented the pretrain–transfer paradigm as a transformative shift in how we approach generative intelligence. From their reasoning capabilities to recent advances in solving complex mathematical problems, LLMs have demonstrated an unprecedented capability and generalizability in capturing intricate patterns across diverse domains (Brown et al., 2020; Achiam et al., 2023; Yang et al., 2025). This paradigm shift raises a compelling question: *Can the foundational principles that enable LLMs to excel at language understanding be extended to solve optimization problems on graphs?*

Graph problems have long been studied in the operations research (OR) community, such as transportation, logistics, and network design. Still, the NP-hard nature of these tasks remains a central bottleneck: exact methods provide optimal solutions at prohibitive cost, whereas heuristics achieve tractability with rare optimality guarantees at the expense of generality (Papadimitriou & Steiglitz, 1981). Recent learning-based efforts fall into two broad strands: (i) problem-specific deep learning (DL) models and (ii) LLM-centric approaches. From pointer networks (Vinyals et al., 2015) and reinforcement learning (RL)-trained attention models to GNN/Transformer variants (Bello et al., 2016; Kool et al., 2018; Kwon et al., 2020) and diffusion solvers, problem-specific DL methods typically rely on specialized decoders/pointer mechanisms (Sun & Yang, 2023) and achieve strong results on synthetic complete graphs. LLM-centric approaches either cast optimization as text generation or use LLMs as modeling/decomposition agents with tool calls (Yang et al., 2023a; Zhang & Luo, 2025; Huang et al., 2024a). However, both strands are trained/evaluated on synthetic complete graphs and

task-specific architectures. The synthetic setting will result in weak feasibility/robustness under real-world constraints; task-specific architectures will be limited in scalability to large instances and exhibit poor cross-task generalization. As a result, they fail to capture transferable structural priors and cannot effectively handle realistic road-network settings where sparsity, geometry, and constraints play a central role.

In this study, we propose the Graph Foundation Model (GFM), a pretraining framework that learns a transferable structural prior over graphs. Analogous to how LLMs learn linguistic priors, GFM is pretrained on large, heterogeneous graph corpora to internalize topological and geometric regularities without using any problem-specific supervision. With this universal structural prior and light task specifications, GFM can solve distance-based graph optimization problems through downstream generation strategies, rather than engineering a solver per task.

Concretely, we transform raw graphs into rich, self-supervised training signals via structure-aware walks inspired by Node2Vec (Grover & Leskovec, 2016). Synthetic trajectories generated from random walks can be used to pretrain the GFM through hidden segment reconstruction, enabling the GFM's interpretability in terms of the graph's connectivity, reachability, and path consistency. Such a pre-training process transfers the concept of words and paragraphs in language to nodes and trajectories under a graph structure, enabling the widely accepted LLM paradigm to be adopted in a graph setting and yielding a task-agnostic backbone that captures reusable graph semantics.

At the stage of inference, GFM functions as a distribution approximator over feasible structures. A simple, task-agnostic decoding, combined with lightweight projection, steers the pretrained prior toward concrete objectives, such as shortest paths, tour variants, and subset selection, while enforcing hard constraints. Crucially, no problem-specific feedback is used during pretraining, in contrast to many neural OR pipelines relying on supervised labels or RL rewards tied to a single problem (Bello et al., 2016). This separation, universal pretraining of the structure, and the subsequent thin task-specific generation, drives strong generalizability across disparate graph optimization tasks on real road networks.

Empirically, our pretrained GFM backbone adapts to diverse distance-based graph optimization tasks, including multiple NP-hard families, without any architectural modification. Across scales from small synthetic graphs ($N=20$) to large real networks ($N=893$), GFM achieves competitive solution quality compared to widely used OR methods such as LKH3 and OR-tools, demonstrating GFM's capability for graph optimization.

In summary, we have the following contributions:

**(1) First graph foundation model for realistic, distance-based graph problems.** We first introduce GFM, a single pretrained model with lightweight constraint projection that excels across multiple graph tasks, spanning from shortest path to NP-hard routing variants, e.g., tour families, on real traffic networks, without any architectural changes.

**(2) Pretraining with random walks and insertion-based reconstruction.** We utilize structure-aware random walks to capture the graph topology and an insertion-based reconstruction curriculum to equip the model with holistic structural priors.

**(3) Transferring the LLM pretrain and adapt paradigm to OR Problems** This work explores a new research direction that transfers the powerful LLM paradigm to OR fields. By demonstrating that the foundational training principles of large language models can be effectively applied to graph-based optimization, we pave the way for leveraging the full spectrum of LLM innovations in graph-based optimization problems.

## 2 RELATED WORK

**Neural Combinatorial Optimization** Neural approaches to combinatorial optimization have evolved from early Hopfield networks (Hopfield & Tank, 1985) to sophisticated architectures leveraging modern deep learning advances. The foundational work of Vinyals et al. (2015) introduced Pointer Networks, demonstrating that sequence-to-sequence models could generate near-optimal Traveling Salesman Problem (TSP) solutions through attention mechanisms. This paradigm was enhanced through RL by Bello et al. (2016) and attention-based encoders by Kool et al. (2018),

who achieved competitive performance on synthetic TSP instances. The POMO framework (Kwon et al., 2020) further improved solution quality through multi-start parallel decoding strategies.

The transformer revolution has significantly advanced neural combinatorial optimization. Bresson & Laurent (2021) achieved remarkable 0.004% optimality gaps on 50-city TSP instances using standard transformer encoders without pointer mechanisms, while Yang et al. (2023b) addressed scalability by reducing attention complexity from $O(n^2)$ to $O(n \log n)$ through Sampled Scaled Dot-Product Attention. More recently, diffusion-based approaches such as DIFUSCO (Sun & Yang, 2023) and general neural CO frameworks like Luo et al. (2023) have extended the paradigm to broader optimization settings. Recent structure-aware approaches like Zhao & Wong (2025) incorporate graph centrality measures and spatial encodings to capture topological relationships. Zhou et al. (2024) proposed to introduce an instance conditional adaptation module in the neural routing solver, enabling the model to adjust according to the features of the input graph dynamically.

However, these methods face critical limitations: they are typically trained on synthetic complete graphs, rely on task-specific architectures that cannot transfer knowledge across different optimization problems, and are fundamentally task-driven rather than structure-driven—designing solvers narrowly around individual problems without cultivating transferable structural priors. Consequently, they struggle to generalize to real-world sparse road networks where sparsity, geometry, and topological constraints are central to optimization objectives.

**Foundation Models for Optimization**  The success of LLM has inspired new paradigms for OR. Direct solving approaches include Yang et al. (2023a)'s OPRO framework, which places LLMs in iterative improvement loops for gradual solution refinement, and Ghimire et al. (2025)'s autoregressive TSP solver that treats tours as token sequences, achieving competitive performance on 100-node instances through Direct Preference Optimization. Multimodal approaches, such as Elhenawy et al. (2024), explore visual reasoning for TSP solving, while evolutionary frameworks, like Liu et al. (2023b), employ LLMs as variation operators within genetic algorithms.

LLM-assisted optimization represents an alternative paradigm where language models handle problem formulation rather than direct solving. Zhang & Luo (2025) developed OR-LLM-Agent for structured prompting and tool use in OR, while Huang et al. (2024a) introduced ORLM for translating natural language descriptions into executable optimization code. Ye et al. (2024) proposed ReEvo, where LLMs generate entire heuristic algorithms through evolutionary search with natural language feedback.

Despite promising results, existing approaches face fundamental challenges that limit their practical applicability. Neural methods trained on synthetic complete graphs struggle with realistic networks where edge connectivity and geometric constraints are paramount. LLM-based approaches suffer from representation mismatches between natural language and graph-theoretic properties, scalability limitations, and an inability to capture the topological and geometric relationships essential for real-world optimization problems. Our work addresses these limitations by developing a foundation model that directly operates on graph structures while preserving the transferability and scalability advantages of the pretrain-transfer paradigm.

## 3 GRAPH FOUNDATION MODEL

Our work aims to instantiate the successful pre-train and fine-tune paradigm (Devlin et al., 2019; Brown et al., 2020) on graph structures. We propose a learnable structural prior to capture the graph's intrinsic topological properties. To achieve this, we devise a self-supervised method based on random walk context reconstruction. We then demonstrate how this powerful prior enables a simple ad-hoc approach to tackle complex OR problems effectively.

### 3.1 METHODOLOGY OVERVIEW

We now introduce the methodology of our GFM, whose overall framework is shown in Figure 1.

Let a weighted graph be $G = (V, E, w)$, and let $\mathcal{Y}(G)$ denote the space of all structured candidates (e.g., paths, tours, subgraphs). The GFM first uses a distance-biased random walk to sample a graph corpus, i.e., a trajectory dataset. We then employ a bidirectional Transformer Encoder to learn a

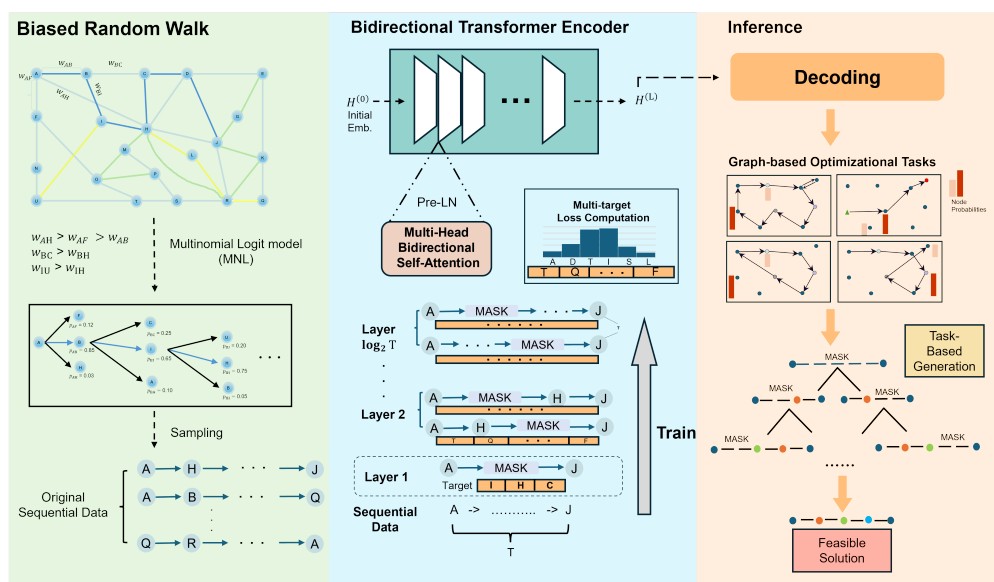

Figure 1: The overall framework of our proposed GFM.

Note. Progressive sample generation with distance-biased random walks provides training trajectories. Node and position embeddings are combined and passed into a Bidirectional Transformer Encoder, which captures graph context without causal masks. The model is optimized through a multi-target reconstruction process and adapted to various graph-based tasks through task-specific generation during inference.

structural prior distribution $\pi(\mathbf{Y} \mid G)$ through an insertion-based trajectory reconstruction process. The learned structural prior $\pi(\mathbf{Y} \mid G)$ is then used to generate required solutions. For any given task specification $s$, the feasible set $\mathcal{F}(G, s) \subseteq \mathcal{Y}(G)$ contains only candidates that satisfy the task constraints, and $J(\mathbf{Y}; G, s)$ denotes the associated objective:

$$\mathbf{Y}^* = \arg \min_{\mathbf{Y} \in \mathcal{F}(G,s)} J(\mathbf{Y}; G, s). \tag{1}$$

Rather than directly solving the exact objective above, our framework employs $\pi(\mathbf{Y} \mid G)$ to generate task-consistent candidates $\mathbf{Y} \in \mathcal{Y}(G)$, by decoding from this prior under feasibility constraints.

## 3.2 Random walk to graph corpora

To establish a training signal for graphs, we transform graph topology into a sequential form through biased random walks. Specifically, given a sparse weighted graph $G = (V, E, w)$, we employ a Node2Vec-style biased random (Grover & Leskovec, 2016) walk with distance-aware utilities to generate path sequences that reflect both topology and metric relations. Formally, starting from node $i = v_t$, the probability of moving to a neighbor $j$ given the previous node $r = v_{t-1}$ is

$$\Pr(v_{t+1} = j \mid v_t = i, v_{t-1} = r) = \frac{\exp\left(-\beta \cdot w_{ij} \cdot b_{p,q}(i, j, r)\right)}{\sum_{k \in \mathcal{N}(i)} \exp\left(-\beta \cdot w_{ik} \cdot b_{p,q}(i, k, r)\right)}, \tag{2}$$

where $w_{ij}$ is the edge weight, $\beta > 0$ is a softmax scaling factor controlling the sharpness of the distribution, and $b_{p,q}(i, j, r)$ denotes the Node2Vec bias term: $p$ penalizes immediate backtracking to the previous node $r$, while $q$ balances the likelihood of exploring outward versus staying local. We introduce a weight bias into the sample utility, allowing the frequency of trajectories to reflect their distance dissimilarities. The process ensures that both distance costs and structural exploration biases are taken into account during walk generation.

## 3.3 Progressive Training Curriculum

To learn the structure prior of the graph, we devise a self-supervised method based on the random walk context reconstruction.

---

**Algorithm 1** Progressive Curriculum Construction from Random Walks

---

**Require:** Random walk $\mathbf{v} = [v_1, \ldots, v_T]$ on graph $G$, max path length $T_{\max}$.
**Ensure:** Training set $\mathcal{D}$ of masked-prediction samples.
 1: Initialize $\mathcal{D} \leftarrow \varnothing$, interior nodes $\mathcal{V}_{\mathrm{int}} = \{v_2, \ldots, v_{T-1}\}$.
 2: Determine number of levels $\ell_{\max} = \min(L_{\max}, \max(L_{\min}, \lfloor \log_2 T \rfloor))$.
 3: **Level 1 (global prior).** Construct $\mathbf{x} = [v_1, \text{MASK}, v_T]$, with admissible targets $\mathcal{Y} = \mathcal{V}_{\mathrm{int}}$. Add
    $(\mathbf{x}, \mathcal{Y}, \ell{=}1)$ to $\mathcal{D}$.
 4: Initialize anchor set $C^{(1)} = [v_1, v_T]$.                      // $C^{(\ell)}$: anchor set at level $\ell$
 5: **for** $\ell = 2$ **to** $\ell_{\max}$ **do**
 6:    Select a gap $(a, b)$ from $C^{(\ell-1)}$ and sample anchor $u$ from $\text{BETWEEN}(a, b \mid \mathbf{v})$ (if empty,
       sample from $\mathcal{V}_{\mathrm{int}}$).
 7:    Insert $u$ into $C^{(\ell-1)}$ to form $C^{(\ell)}$.                      // refine anchors
 8:    **for all** gaps $(a', b')$ in $C^{(\ell)}$ **do**
 9:       Form query $\mathbf{x}$ by inserting MASK between $(a', b')$.
10:       Admissible targets $\mathcal{Y} = \text{BETWEEN}(a', b' \mid \mathbf{v})$.
11:       Add $(\mathbf{x}, \mathcal{Y}, \ell)$ to $\mathcal{D}$.
12:    **end for**
13:    Optionally mask additional random interior positions of $C^{(\ell)}$ and generate samples analo-
       gously.
14: **end for**
15: **return** $\mathcal{D}$

---

We adopt a hierarchical training curriculum that gradually refines supervision from coarse global signals to fine-grained local constraints. The purpose of this curriculum is to enable the model to learn the path context reconstruction. At the first level ($k = 1$), only the endpoints are observed while the entire interior is masked,

$$\ell^1 = [v_1, \text{MASK}, v_T], \quad \mathcal{Y} = \{v_2, \ldots, v_{T-1}\}, \tag{3}$$

$\ell^1$ is the first level of the curriculum $\mathcal{Y}$ is the target set within the walk. This encourages the model to establish a global prior over feasible paths.

At higher levels ($k > 1$), a subset of interior nodes $a_1, \ldots, a_m$ are revealed as anchors. The masked prediction is then restricted to the sub-path between consecutive anchors. For each adjacent anchor pair $(a_j, a_{j+1})$ with $j = 0, 1, \ldots, m-1$, we construct one masked sequence, with admissible target set $\mathcal{Y}_j^k = \{ v \in \mathbf{v} \mid a_j \prec v \prec a_{j+1} \}$,

$$\ell^k = [v_1, a_1, \ldots, a_j, \text{MASK}, a_{j+1}, \ldots, v_T], \tag{4}$$

where $\prec$ denotes the ordering along the original walk. Hence, the level-$k$ training set is

$$\mathcal{L}^k = \{ (\ell_j^k, \mathcal{Y}_j^k) \mid j = 0, \ldots, k-1 \}. \tag{5}$$

To learn from the path completions, we optimize a multi-target loss. For a masked query $x$ with admissible targets set $\mathcal{Y}_s$, let $z \in \mathbb{R}^{|V|}$ be the logits and $P = \text{softmax}(z)$. The loss aggregates probability mass over all valid targets:

$$\ell_{\mathrm{MT}}(s) = -\log \sum_{y \in \mathcal{Y}_s} P_y, \tag{6}$$

The final objective averages the multi-target loss across a mini-batch $\mathcal{B}$:

$$\mathcal{L}_{\mathrm{prog}} = \frac{1}{|\mathcal{B}|} \sum_{x \in \mathcal{B}} \alpha_{\ell(x)} \, \ell_{\mathrm{MT}}(x), \tag{7}$$

where $x$ indexes a query, $\ell(q)$ is its curriculum level, and $\alpha_\ell$ is a level-dependent weight increasing with $\ell$.

### 3.4 MODEL ARCHITECTURE

Our backbone is an encoder-only, pre-norm Transformer (Vaswani et al., 2017), aligned with BERT-style models (Devlin et al., 2019). Each block applies Layer Normalization, multi-head self-attention, and a feed-forward MLP with residual connections. Importantly, our self-attention is

bidirectional, allowing every token to access both its left and right context. The only masking applied is padding, which ensures that computation ignores padded positions while preserving global context.

Formally, let $H \in \mathbb{R}^{T \times d}$ denote the hidden states at the input of a layer ($T$: sequence length, $d$: embedding dimension). Multi-head self-attention computes

$$\text{MHA}(H) = \left[ \text{softmax}\left( \frac{(HW_m^Q)(HW_m^K)^\top}{\sqrt{d_h}} + M_{\text{pad}} \right)(HW_m^V) \right]_{m=1}^{M} W^O, \tag{8}$$

where $M$ is the number of heads, $d_h = d/M$ is the head dimension, $W_m^Q, W_m^K, W_m^V \in \mathbb{R}^{d \times d_h}$ are the learned projection matrices for queries, keys, and values, and $W^O \in \mathbb{R}^{M d_h \times d}$ is the output projection. Here $Q_m = HW_m^Q$, $K_m = HW_m^K$, and $V_m = HW_m^V$, i.e., the input hidden states $H$ are linearly projected into query, key, and value spaces. The notation $[\cdot]_{m=1}^{M}$ denotes concatenation over all heads. Finally, $M_{\text{pad}} \in \mathbb{R}^{T \times T}$ is an additive mask with 0 for valid tokens and $-\infty$ for padding. This bidirectional design enables the model to leverage full-sequence dependencies for predicting masked nodes.

### 3.5 DECODING FOR GRAPH OPTIMIZATION

At inference time, we obtain task-specific solutions by decoding from the learned structural prior under feasibility constraints. Formally, the final solution is generated as

$$\widehat{\mathbf{Y}} = \text{Decode}\big(\pi(\cdot \mid G), s\big), \tag{9}$$

where $\widehat{\mathbf{Y}}$ denotes the final solution generated by our model and $\pi(\cdot \mid G)$ provides the structural bias learned from the graph $G$, $s$ specifies task-dependent constraints (e.g., source/target in shortest path, required nodes in tours), and $\text{Decode}(\cdot)$ denotes our decoding strategy that maps the prior distribution into a feasible solution.

## 4 EXPERIMENTS

We evaluate GFM on one synthetic graph and two real-world road networks across four problems: shortest path (SP) and three NP-hard tasks—Graphic Traveling Salesman Problem (Graphic-TSP), tour problem with the same origin and destination, and tour problem with different origins and destinations (Martin et al., 2022). The results highlight GFM's cross-task capability and high-quality solving performance on graph optimization problems.

### 4.1 DATASETS AND PROBLEM CONSTRUCTION

**Datasets**   To ground our evaluation on reproducible yet realistic scenarios, we adopt one controlled simulation graph and two real-world road networks, all are undirected connected graphs:

**(1) Simulation graph** ($N$=20). A randomly generated graph with 20 nodes and 34 edges. The graph has a density of 0.1789 and an average degree of 3.40.

**(2) Chengdu–Longquanyi** ($N$=132). A real road network from Longquanyi District, Chengdu, China, with 132 nodes and 222 edges. The graph has a density of 0.0257 and an average degree of 3.36. Original edge lengths are in meters (m) and converted to kilometers (km) for normalization.

**(3) Berkeley** ($N$=893). A real road network from Berkeley, CA with 893 nodes and 1413 edges, density 0.0035, and average degree 3.16. The graph is obtained via `OSMnx` v2.0+ by clipping a 1.3 km radius around the landmark "Downtown Berkeley BART."

**Problem Construction**   Our goal is to achieve a city-scale GFM that is evaluated on real-world road graphs. Accordingly, we instantiate four distance-driven tasks on the graph $G = (V, E, w)$:

**(1) Shortest Path (SP).** Given $s, t \in V$, find a minimum-length $s{\to}t$ path under edge lengths $w$.

**(2) Graphic Traveling Salesman Problem (Graphic-TSP).** Given a required node set $R \subseteq V$, find a minimum-length closed walk that visits all $r \in R$ on the road graph. Unlike metric TSP on complete graphs, Graphic-TSP on general graphs naturally permits vertex/edge repetitions when necessary,

which matches urban networks with cul-de-sacs and limited connectivity, see approximation results and discussion in Sebö & Vygen (2012).

**(3) Tour Problem with Same Origin and Destination (TP-SOD)** Given a start/end depot $o \in V$ and a required/attractive POI set $R$, produce a feasible tour starting and ending at $o$ that visits all $R$ while minimizing travel length. This is a closed special case of the Orienteering/Prize-Collecting family on road networks, widely studied in the orienteering literature and its variants (Gunawan et al., 2016). It falls under the broad class of tour problems (Martin et al., 2022), where the aim is to plan efficient sightseeing tours for visitors (Vansteenwegen et al., 2011).

**(4) Tour Problem with Different Origin and Destination (TP-DOD).** Given distinct $o, d \in V$ and a required POI set $R$, find a minimum-length open $o{\to}d$ walk that visits all $R$. Similar to TP-SOD, this also belongs to the family of tourist trip problems, modeling practical scenarios such as day trips between different locations (Vansteenwegen et al., 2011; Martin et al., 2022).

## 4.2 EXPERIMENTAL SETTINGS

**Hyperparameters** Our Graph Foundation Model (GFM) adopts Transformer backbones with task-specific settings. On the simulation graph ($N{=}20$), we use a 6-layer, 6-head encoder ($d{=}192$, dropout 0.1, batch size 64, seq. length 10) trained for 15k steps with Adam ($5{\times}10^{-4}$). For Chengdu ($N{=}132$), we employ an 8-layer, 8-head encoder ($d{=}256$, batch size 32, seq. length 20) trained on 2.73M walks for 15k steps at $3 \times 10^{-4}$. For Berkeley ($N{=}893$), we use a 6-layer, 6-head encoder ($d{=}192$, seq. length 200, batch size 64) for 150k steps at $5 \times 10^{-4}$. Parameter counts range 2.7M–6.7M. Baselines follow recommended settings: LKH3 with MAX_TRIALS=10,000 and RUNS=10; OR-Tools with PATH_CHEAPEST_ARC + GUIDED_LOCAL_SEARCH; Gurobi with gaps 0.1–5% and limits 30–1800s; A* with $c_{\text{puct}} = 1.4$, $\alpha = 1.8$. Further details are in Appendix B and D.

**Runtime** All experiments were run on an Intel i5-12400F CPU, 32GB RAM, and RTX 4060 Ti (8GB) with CUDA 12.9. Classical solvers (Dijkstra, OR-Tools, LKH3, Gurobi) utilized official implementations, while LLM baselines (ChatGPT5, OR-LLM-Agent, Qwen3-235B) were accessed via APIs. GFM was trained and tested locally under the same environment, ensuring timing comparability across all methods. Detailed runtime protocols are in Appendix D.

**Baselines** We compare our approach against a broad set of baselines that cover exact solvers, classical heuristics, and modern learning-based approaches. For the shortest path (SP), we include the label-setting algorithm of Dijkstra (1959) as the ground-truth optimum, together with the A* heuristic search and Google's OR-Tools implementation. For tour-based problems, we utilize the state-of-the-art Lin–Kernighan–Helsgaun framework (LKH3) (Helsgaun, 2017), the Christofides approximation followed by 2OPT local refinement, and simple greedy strategies, such as Nearest Neighbor (NN). We also report Gurobi as a commercial MILP solver under fixed time limits, serving as a near-exact reference for NP-hard cases. On the learning side, we benchmark the Attention Model (AM) of Kool et al. (2018), an RL-trained neural combinatorial optimization architecture, under both greedy and sampling-based decoding, and the Instance-Conditioned Adaptation framework (ICAM) (Zhou et al., 2024), which adapts neural solvers to different instance scales via lightweight conditioning modules. We further evaluate OR-LLM-Agent (Zhang & Luo, 2025), which leverages structured prompting and tool use, and two large language models—ChatGPT5 and Qwen3-235B (Yang et al., 2025)—both in zero-shot settings with JSON-constrained outputs. Finally, we compare our proposed Graph Foundation Model (GFM) with this approach, which integrates curriculum-pretrained structural priors and task-specific decoding.

## 4.3 MAIN RESULTS

**Evaluation Metrics** We primarily report three metrics across all tasks: (**Obj**) the objective value, defined as the total path or tour length; (**Succ**) the success rate, i.e., the fraction of outputs that satisfy all task-specific feasibility constraints; and (**Time**) the average runtime per instance (s).

**Result Discussion** Table 1 reports results on simulation, Chengdu, and Berkeley networks. GFM achieves near-optimal objective values with almost perfect feasibility across all tasks.

Table 1: Performance across four routing problems on **Simulation** ($N = 20$), **Chengdu** ($N = 132$), and **Berkeley** ($N = 893$). Entries report **objective (km)**, **success (%)**, and **time (s)**.

| | Method | Simulation ($N = 20$) | | | Chengdu ($N = 132$) | | | Berkeley ($N = 893$) | | |
|---|---|---|---|---|---|---|---|---|---|---|
| | | Obj (km) | Succ (%) | Time (s) | Obj (km) | Succ (%) | Time (s) | Obj (km) | Succ (%) | Time (s) |
| **SP** | Dijkstra | 6.09 | 100% | 0.000 | 5.09 | 100% | 0.000 | 2.23 | 100% | 0.002 |
| | A* | 6.09 | 100% | 0.000 | 5.09 | 100% | 0.000 | 2.23 | 100% | 0.002 |
| | OR-Tools | 6.09 | 100% | 0.000 | 5.09 | 100% | 0.000 | 2.23 | 100% | 0.005 |
| | OR-LLM-Agent | 15.22 | 30% | 19.800 | – | 0% | 45.100 | – | 0% | 91.700 |
| | ChatGPT5 | 8.23 | 99% | 3.394 | 8.85 | 62% | 3.720 | – | 0% | |
| | Qwen3-235B | 9.81 | 97% | 1.89 | 11.98 | 45% | 3.050 | – | 0% | 3.820 |
| | **GFM** | **6.24** | **99%** | **0.046** | **5.19** | **100%** | **0.821** | **2.43** | **98%** | **23.109** |
| **Graphic-TSP** | LKH3 | 39.48 | 100% | 0.016 | 87.50 | 100% | 2.020 | 84.75 | 100% | 2328.000 |
| | Gurobi(30min) | 39.48 | 100% | 0.147 | 90.64 | 100% | 46.423 | 120.00 | 100% | 1800.000 |
| | OR-Tools | 39.48 | 100% | 15.000 | 88.12 | 100% | 30.000 | 109.78 | 100% | 690.000 |
| | Chr. + 2OPT | 42.50 | 100% | 0.007 | 97.78 | 100% | 20.030 | 92.86 | 100% | 606.502 |
| | Greedy | 44.43 | 100% | 0.001 | 111.51 | 100% | 0.006 | 113.76 | 100% | 0.213 |
| | OR-LLM-Agent | – | 0% | 246.830 | – | 0% | 45.100 | – | 0% | 16.530 |
| | AM (greedy) | 100.91 | 100% | 0.042 | 104.53 | 100% | 0.020 | – | – | – |
| | AM (10) | 82.16 | 100% | 0.054 | 104.31 | 100% | 0.017 | – | – | – |
| | AM (100) | 74.83 | 100% | 0.160 | 101.35 | 100% | 0.017 | – | – | – |
| | AM (1000) | 70.83 | 100% | 1.366 | 100.36 | 100% | 0.018 | – | – | – |
| | ICAM | 41.14 | 100% | 0.159 | 103.25 | 100% | 1.982 | 122.51 | 100% | 571.172 |
| | Qwen3-235B | 103.95 | 41% | 2.690 | – | 0% | 22.410 | – | 0% | 16.530 |
| | **GFM** | **39.48** | **100%** | **0.381** | **99.00** | **100%** | **6.405** | **100.87** | **100%** | **129.58** |
| **TP-SOD** | LKH3 | 23.96 | 100% | 0.003 | 22.55 | 100% | 0.003 | 8.77 | 100% | 0.042 |
| | Gurobi(30s) | 23.96 | 100% | 0.007 | 22.55 | 100% | 0.021 | 8.77 | 100% | 0.038 |
| | OR-Tools | 23.96 | 100% | 10.000 | 22.55 | 100% | 10.000 | 8.77 | 100% | 10.000 |
| | OR-LLM-Agent | 20.10 | 76% | 20.100 | – | 3% | 72.700 | – | 0% | 47.000 |
| | AM (greedy) | 30.45 | 100% | 0.033 | 27.78 | 100% | 0.004 | – | – | – |
| | AM (10) | 30.45 | 100% | 0.039 | 25.46 | 100% | 0.003 | – | – | – |
| | AM (100) | 30.62 | 100% | 0.040 | 25.20 | 100% | 0.003 | – | – | – |
| | AM (1000) | 30.62 | 100% | 0.039 | 25.20 | 100% | 0.003 | – | – | – |
| | Qwen3-235B | 58.83 | 80.2% | 2.656 | – | 0% | 2.838 | – | 0% | 9.830 |
| | **GFM** | **24.72** | **100%** | **0.03** | **20.31** | **100%** | **0.071** | **8.82** | **100%** | **0.077** |
| **TP-DOD** | LKH3 | 22.01 | 100% | 0.002 | 19.99 | 100% | 0.003 | 7.84 | 100% | 0.061 |
| | Gurobi(30s) | 22.01 | 100% | 0.004 | 19.93 | 100% | 0.014 | 7.84 | 100% | 0.064 |
| | OR-Tools | 22.01 | 100% | 10.003 | 19.99 | 100% | 10.006 | 7.99 | 100% | 9.958 |
| | OR-LLM-Agent | 21.40 | 40% | 21.400 | – | 1% | 68.300 | – | 0% | 61.200 |
| | AM (greedy) | 31.24 | 100% | 0.002 | 27.67 | 100% | 0.004 | – | – | – |
| | Qwen3-235B | 38.04 | 41% | 1.750 | – | 0% | 8.305 | – | 0% | 10.000 |
| | **GFM** | **22.87** | **100%** | **0.037** | **20.31** | **100%** | **0.068** | **7.84** | **100%** | **0.104** |

*Note.* "–" has two meanings: when the **success rate** is 0%, the method produced outputs but none satisfied task constraints; when all three entries are "–", the method could not generate solutions for that problem setting at all.

Compared to classical solvers, GFM provides competitive solutions with higher efficiency on large-scale problems. This efficiency advantage is most pronounced when solving Graphic-TSP problems. As the network size increases, the runtime of solver-based methods grows significantly, ranging from 606 to 2,328 seconds. In contrast, our GFM-based approach requires only 129 seconds on the same large-scale problems, while delivering solutions of comparable quality (100.87 km versus 84.75-120 km). This advantage stems from the fundamental mechanistic difference between the two approaches. Solver-based methods often rely on near-exhaustive search techniques to find high-quality solutions, which can be computationally intensive. GFM, however, generates solutions directly by leveraging a deep understanding of the graph structure. Our experiments validate the efficiency and effectiveness of the GFM approach.

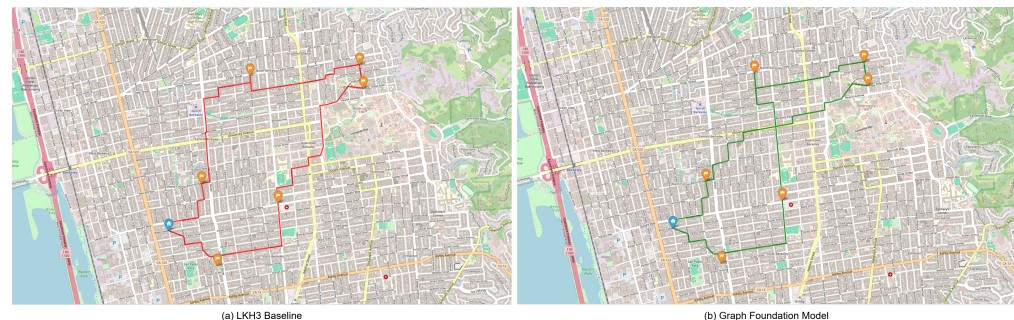

(a) LKH3 Baseline                                    (b) Graph Foundation Model

Figure 2: A comparative example of TP-SOD Solutions on the Berkeley road network

Note. GFM leverages learned structural priors to better capture the geometry and connectivity of real road networks, producing competitive solutions that demonstrate a deep understanding of graph structure.

Furthermore, we visualized the paths generated by LKH3 and GFM for the Tour-SOD problem on the Berkeley network, as shown in Figure 2. This visualization provides further evidence of the high quality of the solutions produced by GFM.

Compared with existing transformer-based methods, GFM can produce solutions with higher quality, e.g, shorter paths and tours. For example, when solving the Graphic-TSP problem on the simulation network, the path length produced by the AM method is approximately twice that of the GFM result. This is because Transformer-based methods are trained on task-specific synthetic graphs. The training setting significantly differs from real road networks that are sparse and topologically constrained, thus limiting their applicability to real mobility systems. In contrast, GFM excels at deeply exploring the network structure and fully perceiving the constraints of the real road network. This advantage enables it to provide higher-quality solutions for real-world graph-based problems.

Compared with current LLM-based baselines, GFM maintains solution validity and quality across different scales and optimization tasks. As network size increases from small to large scale, their success rate drops sharply (e.g., ChatGPT5 in the SP problem: $99\%(N = 20)$, $62\%(N = 132)$, and $0\%(N=893)$). This infeasibility result highlights the inherent limitations of LLMs in directly or indirectly solving graph optimization problems. This discrepancy arises because natural language, which LLMs excel at processing, possesses a degree of inherent stochasticity. In contrast, graph optimization problems are governed by strict structural constraints and explicit objectives, a paradigm that falls outside the domain of what LLMs are adept at handling. Since GFM is trained to capture the network's structure and connectivity, it consistently preserves solution quality as the graph size scales.

These results demonstrate the strength of our GFM on solution quality, efficiency, and stability, which validates the application potential of our GFM framework.

## 5   CONCLUSION

We presented the **Graph Foundation Model (GFM)**, which extends the pretrain–adapt paradigm of large language models to OR problems on graphs. By pretraining on structure-aware random walks with reconstructed trajectories training, GFM internalizes transferable structural priors that capture the geometry and topology of real road networks. This universal backbone enables lightweight solution generation across multiple NP-hard routing families, achieving competitive performance while maintaining efficiency. Our findings highlight the promise of foundation models for graphs: pretraining once on diverse networks can provide a reusable structural prior that generalizes broadly across optimization tasks, marking a step toward universal foundation-level solvers in operations research.

# 6    ETHICS STATEMENT

This work adheres to the ICLR Code of Ethics. In this study, no human subjects or animal experimentation was involved. All datasets used, including the dataset derived from OSMnx, were collected in compliance with relevant usage guidelines and properly cited, ensuring no violation of privacy. We have taken care to avoid any biases or discriminatory outcomes in our research process. No personally identifiable information was used, and no experiments were conducted that could raise privacy or security concerns. We are committed to maintaining transparency and integrity throughout the research process.

# 7    REPRODUCIBILITY STATEMENT

We have made every effort to ensure that the results presented in this paper are reproducible. Upon acceptance, we will release all code and datasets in a public repository, together with a full description of our contributions, experimental setup, model configurations, and hardware details. This will facilitate replication and verification of our work. We believe these measures will enable other researchers to reproduce our results and build upon our approach.

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

# A  DATA PROCESSING AND VISUALIZATION

## A.1  ROAD-GRAPH CONSTRUCTION

**Datasets.** To ground our evaluation on reproducible yet realistic scenarios, we adopt one controlled simulation graph and two real-world road networks.

**Simulation graph** ($N{=}20$)**.** To emulate traffic properties while ensuring reproducibility, we construct a fixed sparse backbone of 20 nodes with several cross links. Edge weights are sampled once with a fixed random seed ($s{=}42$): "arterial" roads from a uniform distribution $U(1.0, 3.0)$ to model lower travel costs, and cross connections from $U(3.5, 7.0)$ to represent less preferred detours. The topology is fixed, with only minor coordinate jitter for visualization, making the instance traffic-informed and exactly reproducible.

**Chengdu–Longquanyi** ($N{=}132$)**.** We extract the road network of Longquanyi District, Chengdu, as a self-collected dataset. The graph is preprocessed into an undirected primal form, and retains geographic coordinates for each node.

**Berkeley** ($N{=}893$)**.** The Berkeley network is obtained from OpenStreetMap using `OSMnx` v2.0+ with `network_type=drive`, which filters for drivable roads. We clip a radius of $1300\,\text{m}$ centered on the landmark "Downtown Berkeley BART," yielding a sparse, connected graph with 893 nodes. Node IDs are remapped to $0, \ldots, N{-}1$, and edge weights are computed as geometric distances (meters divided by 1000). All parameters (OSMnx version, query strings, radius, preprocessing) are fixed in our scripts to guarantee reproducibility. Raw OSM MultiDiGraphs are simplified into undirected simple graphs: we remove self-loops and duplicate edges, retain the largest connected component, and remap node IDs to $0..(N{-}1)$ for efficient embedding. Edge weights are set to geometric lengths (meters) converted to kilometers, i.e., `length_km = length_m/1000`. Each node stores its GPS coordinates $(\text{lon}, \text{lat})$ for visualization and for computing path lengths consistent with our evaluation objective (km).

# B  BASELINE ADAPTATION ON SPARSE ROAD NETWORKS

Most classical solvers and some baselines (e.g., LKH3 (Helsgaun, 2017), OR-Tools, Gurobi, ICAM Zhou et al. (2024)) assume a complete graph input, where every pair of nodes has a direct edge. However, real road networks are sparse: many node pairs are not directly connected. To enable fair evaluation, we adopt two adaptation strategies: metric closure and virtual coordinate embedding.

**Metric closure.** Let $G = (V, E, w)$ be the original weighted road network. For any pair $u, v \in V$, define
$$d(u, v) \;=\; \min_{p \in \mathcal{P}(u,v)} \sum_{(i,j) \in p} w(i, j),$$
where $\mathcal{P}(u, v)$ is the set of all paths between $u$ and $v$. The metric closure is the complete graph $G^* = (V, E^*, d)$ with edge weights equal to shortest-path distances in $G$.

**Virtual edge construction.** For path problems with distinct start and end nodes, e.g., TP-DOD, solvers like LKH3 require a closed TSP formulation. We therefore introduce a virtual edge from the terminal node back to the start node with weight zero, transforming the open-path problem into a tour. After solving, this artificial edge is removed and the remaining sequence is expanded back into a feasible path in $G$.

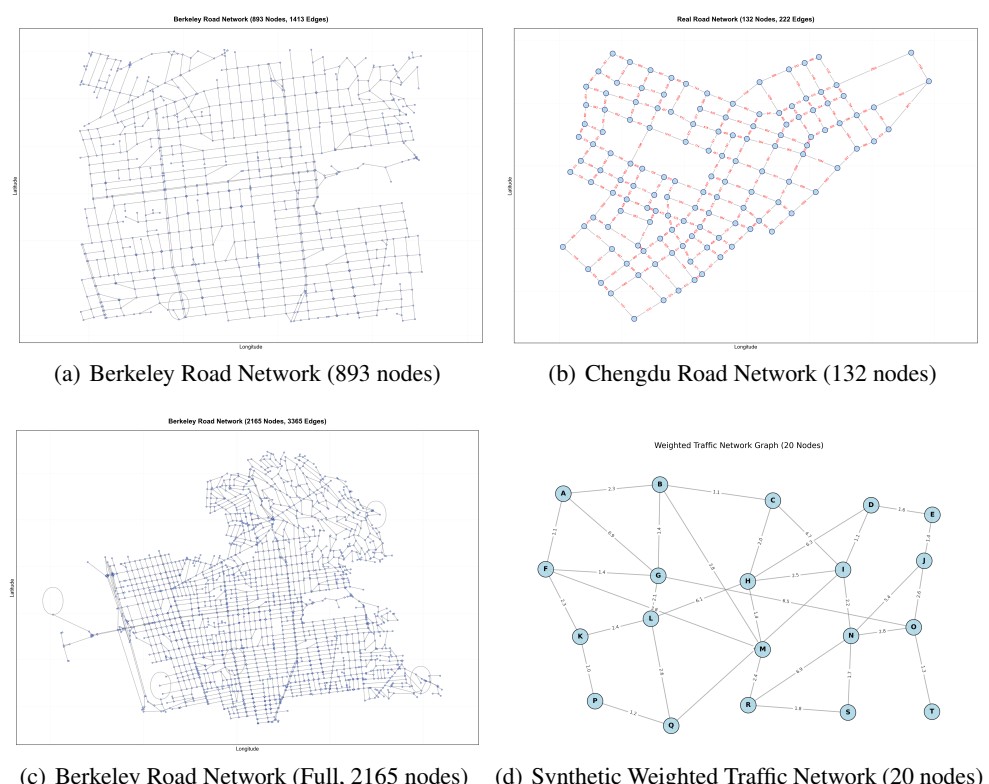

(a) Berkeley Road Network (893 nodes)  (b) Chengdu Road Network (132 nodes)

(c) Berkeley Road Network (Full, 2165 nodes)  (d) Synthetic Weighted Traffic Network (20 nodes)

Figure 3: Examples of road network datasets used in our experiments. Real-world road networks are acquired from OpenStreetMap (**Berkeley**), while a synthetic traffic network provides controlled small-scale settings.

**Application.** For both strategies, the adapted representation (closure graph $G^*$ or virtual edge construction) is fed to the baseline solver (e.g., LKH3, OR-Tools, AM, ICAM) to generate a high-level tour $\pi^*$. Each edge $(u, v) \in \pi^*$ is then expanded back into its shortest path in the original graph $G$, guaranteeing feasibility while preserving the solver's optimization logic.

## C  RELATED WORKS (EXTENDED)

**Graph Neural Networks**  The application of neural networks to combinatorial optimization can be traced back to the pioneering work of Hopfield & Tank (1985), who first employed Hopfield networks to solve the traveling salesman problem, establishing the foundation for using neural network dynamics in NP-hard optimization problems. Early graph neural network variants, including GCN Kipf & Welling (2016), GAT Velickovic et al. (2017), and GraphSAGE Hamilton et al. (2017), demonstrated significant success in learning relational data and solving graph-based problems.

The paradigm shift toward sequence-to-sequence models in combinatorial optimization began with Pointer Networks Vinyals et al. (2015), which utilized RNNs to encode city coordinates and employed attention mechanisms during decoding to directly generate city visiting sequences. This work established that end-to-end neural networks could learn to generate near-optimal TSP solutions without explicit algorithmic guidance. Building upon this foundation, Bello et al. (2016) introduced reinforcement learning with actor-critic methods to train pointer networks using negative tour distances as rewards, demonstrating that solution quality could be improved without supervised optimal solutions.

The integration of attention mechanisms further advanced the field when Kool et al. (2018) combined attention-based city encoding with pointer network selection mechanisms, training the resulting model through reinforcement learning. This was extended by Kwon et al. (2020), who proposed

POMO (Policy Optimization with Multiple Optima), employing multi-start parallel decoding to generate multiple solutions and improve optimization quality. Concurrently, Joshi et al. (2019) applied graph neural networks directly to TSP by using graph convolutions to encode complete graph structures combined with greedy selection strategies.

**Transformer-Based Approaches**   Following the transformer revolution  Vaswani et al. (2017), several works have adapted transformer architectures for combinatorial optimization.  Bresson & Laurent (2021) proposed using standard transformer encoders without pointer mechanisms, encoding city coordinate sequences as input tokens and outputting city-sorted logits representing visit orders through reinforcement learning optimization.  Their implementation achieved remarkable performance with optimality gaps of 0.004% on 50-city instances and approximately 0.37% on 100-city problems after adequate training and beam search.

However, the quadratic complexity of standard attention mechanisms poses scalability challenges for large-scale problems. To address this limitation, recent works have focused on efficiency improvements.  Yang et al. (2023b) introduced TSPformer with Sampled Scaled Dot-Product Attention (SSDPA), reducing the standard attention complexity from $O(n^2)$ to $O(n \log n)$ while maintaining competitive performance on 1000-node TSP instances with minimal accuracy loss. Similarly,  Jin et al. (2023) proposed Pointerformer, incorporating multi-pointer decoders and data augmentation techniques leveraging the rotation and reflection invariant properties of TSP solutions.

Alternative approaches have explored non-autoregressive formulations.  Khalil et al. (2017) trained graph neural networks in a supervised manner to directly output tours as adjacency matrices, converting results to feasible solutions via beam search.  However, this non-autoregressive approach achieved only 2.7% optimality gap for n=20, underperforming compared to autoregressive methods.  Deudon et al. (2018) developed a transformer-based model outputting fractional solutions to multiple TSP variants, treating results as linear relaxation solutions and employing beam search for integer feasibility.

Most recently,  Zhao & Wong (2025) designed a structure-aware transformer incorporating closeness centrality rather than degree centrality for node representation, integrating spatial distance encoding to capture inter-node relationships.  Their improved pointer mechanism incorporates information from all visited nodes into the context state, capturing the dynamic evolution of path construction and outperforming classical heuristics, benchmark methods, and previous learning-based approaches across diverse test scenarios. Also,  Zhou et al. (2024) incor- porate graph centrality measures and spatial encodings to capture topological relationships.

**Large Language Models for Combinatorial Optimization**   The emergence of large language models has opened new avenues for approaching combinatorial optimization problems.  Recent research in this domain can be categorized into two primary paradigms:  direct problem solving through language models and LLM-assisted optimization frameworks.

In the direct solving paradigm, several works have explored applying LLMs to graph problems through various architectural adaptations.  Tang et al. (2023) developed GraphGPT, which embeds graph structures through graph encoders and aligns them with textual information, enabling simultaneous utilization of semantic and structural knowledge in graph tasks.  Liu et al. (2023a) proposed OFA, a unified framework that integrates graph tasks across different domains through Text Attribute Graphs and Graph Prompt Paradigms, achieving cross-task generalization capabilities.

For TSP-specific applications,  Yang et al. (2023a) introduced OPRO (Optimization through Prompts), which places LLMs in an iterative improvement loop where existing candidate solutions and their costs enable the generation of better solutions. While not guaranteeing optimality, this approach demonstrates the potential of language generation with feedback for gradual route improvement.  Liu et al. (2023b) proposed LLM-driven evolutionary algorithms (LMEA), where LLMs serve as variation operators within evolutionary frameworks, selecting parent solutions and performing crossover/mutation operations through textual outputs. Although limited to small instances (up to 20 nodes), LMEA achieved competitive performance with traditional TSP heuristics.

Recent advances have explored more sophisticated approaches.  Ghimire et al. (2025) developed a one-shot autoregressive TSP solver that treats tours as sequences of city tokens, training the model to predict tours autoregressively and applying Direct Preference Optimization for quality improvement.

This approach achieved tours within a few percent of optimal length on instances up to 100 nodes. Elhenawy et al. (2024) investigated visual reasoning with multimodal LLMs, where TSP instances are presented as images and the LLM reasons about efficient tours through visual intuition, achieving competitive performance with OR-Tools on smaller instances.

The second paradigm focuses on LLM-assisted optimization frameworks that leverage language models for problem formulation and modeling. Zhang & Luo (2025) developed OR-LLM-Agent, an agentic framework that models and solves operations research problems through structured prompting and tool use. Huang et al. (2024a) introduced ORLM, which trains specialized LLMs for optimization modeling, enabling the translation of natural language problem descriptions into formal models or executable code for traditional solvers.

Huang et al. (2024b) examined LLMs for vehicle routing problems described in natural language, demonstrating that GPT-4 can serve as a coding assistant for combinatorial solvers with self-debugging and refinement capabilities. Ye et al. (2024) proposed ReEvo (Reflective Evolution), where LLMs generate entire heuristic algorithms rather than individual solutions, employing evolutionary strategies with natural language feedback to evolve competitive algorithms across multiple NP-hard problems.

Despite these advances, LLM-based approaches face fundamental limitations including scalability constraints, the mismatch between natural language representations and graph-theoretic constraints, and the inability to capture geometric and topological properties essential for real-world networks. These challenges motivate the need for more principled approaches that bridge language modeling paradigms with graph optimization requirements.

## D  IMPLEMENTATION DETAILS

**Hyperparameters**  Our Graph Foundation Model (GFM) adopts Transformer backbones with task-specific configurations across three datasets. For the simulation graph ($N{=}20$), we use a 6-layer, 6-head encoder with embedding dimension $d{=}192$, dropout 0.1, batch size 64, and sequence length 10, trained for 15k steps with Adam (learning rate $5 \times 10^{-4}$). The Chengdu road graph ($N{=}132$) employs a larger 8-layer, 8-head encoder with $d{=}256$, dropout 0.1, batch size 32, and sequence length 15, trained on 2.73M random walks for 15k steps at $3 \times 10^{-4}$ learning rate. The Berkeley road graph ($N{=}893$) uses a 6-layer, 6-head encoder with $d{=}192$, extended sequence length 200 to match longer walks, batch size 64, and a total of 150k training steps at $5 \times 10^{-4}$. Parameter counts range from 2.7M–6.7M, with Transformer blocks contributing over 95% of weights. For classical solvers and baselines, necessary adaptations such as metric closure were applied to ensure applicability on sparse road graphs; Adaptation details are provided in Appendix B. Baselines follow their recommended hyperparameter settings to ensure comparability. For LKH3 we set `MAX_TRIALS`=10,000, `RUNS`=10, and fixed random seeds, consistent with Helsgaun's implementation (Helsgaun, 2017). OR-Tools solvers adopt `PATH_CHEAPEST_ARC` as first solution strategy and `GUIDED_LOCAL_SEARCH` for improvement, with a 300s cap for large graphs. Gurobi is tuned with gap tolerances from 0.1%–5%, heuristic strength 0.2–0.5, and runtime limits of 30–1800s depending on problem size. A* search uses $c_{\text{puct}} = 1.4$, $\alpha = 1.8$, and pruning optimizations (lower-bound cuts, transposition tables, progressive widening). Classical heuristics such as Nearest Neighbor and Greedy follow standard greedy insertion without additional tuning, while attention-based neural baselines adopt the publicly released settings of Kool et al. (Kool et al., 2018). These consistent hyperparameter choices ensure fairness across solvers and facilitate reproducibility.

**Runtime**  All experiments were executed on a single workstation equipped with an Intel Core i5-12400F CPU (6 physical cores, 12 threads), 32GB DDR4 RAM, and an NVIDIA RTX 4060 Ti GPU with 8GB VRAM, running CUDA 12.9 and driver version 576.28. GPU utilization during training averaged below 20%, with peak memory footprint under 2GB, ensuring that models fit comfortably within consumer-grade hardware. To ensure fairness, each baseline solver was executed under identical hardware constraints and with fixed random seeds. Classical methods (Dijkstra, OR-Tools, LKH3, Gurobi) were executed using their official high-performance implementations with optimized C/C++ backends. LLM-based methods were evaluated through standardized APIs: ChatGPT5 and OR-LLM-Agent via the OpenAI API, and Qwen3-235B via the Together AI API.

GFM training and inference were conducted locally under the same environment. All timing results are therefore directly comparable across methods under uniform runtime protocols.

# E   LLM USAGE

Large Language Models (LLMs) were used to aid in the writing and polishing of the manuscript. Specifically, we used an LLM to assist in refining the language, improving readability, and ensuring clarity in various sections of the paper. The model helped with tasks such as sentence rephrasing, grammar checking, and enhancing the overall flow of the text. Separately, we also employed some LLMs purely as baselines in our experiments, serving only as comparative references against our proposed method.

It is important to note that LLMs were not involved in the ideation, research methodology, or experimental design. All research concepts, ideas, and analyses were developed and conducted by the authors. The authors take full responsibility for the content of the manuscript, including any text generated or polished by the LLM. We have ensured that the LLM-assisted text adheres to ethical guidelines and does not contribute to plagiarism or scientific misconduct.

