# OpenReview forum: "Graph Foundation Models: Bridging Language Model Paradigms and Graph Optimization"
_ICLR.cc/2026/Conference — Submitted to ICLR 2026_

### Official Review · Reviewer_GvNj · 2025-10-23

**Soundness:** 2
**Presentation:** 2
**Contribution:** 2
**Rating:** 4
**Confidence:** 3

**Summary:**

The paper proposes the Graph Foundation Model (GFM), a framework that transfers the pretrain–transfer paradigm of large language models (LLMs) to graph-structured data for operations research (OR) and graph optimization problems.

**Strengths:**

1. This work clearly defines the model architecture, training curriculum, and decoding strategies for constrained graph tasks.

2. It introduces a foundation model perspective for graph optimization, moving beyond task-specific architectures.

3. It evaluates across multiple scales and graph problems (SP, Graphic-TSP, TP-SOD, TP-DOD)

**Weaknesses:**

1. Reported scalability only reaches ~900 nodes, and further validation or discussion on larger networks is needed.

2. It’s unclear if performance gains arise primarily from architectural differences or training scale.

3. The quality of the diagram (Fig. 1) is poor, in which the font size is too small to read. And the logic is also not good to deliver the key ideas.

4. GFM is a big concept. As this work is limited only to distance-related problems, GFM is overstated.

5. No ablation and parameter sensitivity studies to isolate/analyze the contribution of each training component.

**Questions:**

1. Can the proposed method be extended to other combinatorial problems such as influence maximization and community detection?

---

> ### Author Response · Authors · 2025-11-26
> **Response to Reviewer GvNj**
>
> **Question:** We thank the reviewer for this insightful comment, which points to downstream tasks that our Graph Foundation Model (GFM) can in fact support but were not explicitly demonstrated in the original submission. Here we provide a concrete instantiation on community detection, and a second one on influence maximization.
>
> For community detection, we use the pretrained GFM encoder to derive node embeddings on the American College Football network. Concretely, for each node $i$, we sample $M$ short MNL-biased random walks of length $L$ starting from every node. Each walk is fed through the encoder to obtain contextual hidden states along the walk. We then collect all positions where the token equals node $i$, and define the node embedding $\mathbf{e}_i$ as the average of its contextual hidden states:
> $$
> \mathbf{e} _ i
> = \frac{1}{|\mathcal{S} _ i|} \sum _ {(m,t)\in\mathcal{S} _ i} H^{(m)} _ {t} \in \mathbb{R}^d,
> $$
> where $\mathcal{S}_i$ is the set of all $(m,t)$ pairs at which node \(i\) appears in the sampled walks, and $H^{(m)}_t$ is the encoder hidden state at step $t$ of walk $m$. Finally, we run standard $k$-means on $\{\mathbf{e} _ i\} _ {i=1}^n$, matching the number of conferences, to obtain community assignments.
>
> **Table 1.** Community detection results on the American College Football network.
>
> | Method                             | Adjusted Rand Index (ARI) | Normalized Mutual Information (NMI) |
> |------------------------------------|---------------------------|-------------------------------------|
> | Louvain (modularity maximization)  | 0.9092                    | 0.9764                              |
> | Label Propagation                  | 0.3686                    | 0.7343                              |
> | Greedy Modularity                  | 0.5301                    | 0.7624                              |
> | DeepWalk + k-means                 | 0.6923                    | 0.8439                              |
> | node2vec + k-means                 | 0.9216                    | 0.9545                              |
> | Adjacency embedding + k-means      | 0.9206                    | 0.9543                              |
> | **GFM (ours)**                     | **0.9618**                | **0.9774**                          |
>
> Beyond community detection, we also instantiate an influence maximization task on the Berkeley road network using the same GFM node embeddings. We construct an embedding-based affinity score between nodes, apply a greedy selection rule under a cardinality constraint to choose \(k\) seed nodes, and then evaluate the selected seeds under an Independent Cascade (IC) model via Monte Carlo simulation.
>
> **Table 2.** Influence maximization performance on the Berkeley road network (expected spread under the IC model).
>
> | \(k\) | Random | Degree | PageRank | Greedy-IC | GFM-IM |
> |------:|:------:|:------:|:--------:|:---------:|:------:|
> | 0     | 0.00   | 0.00   | 0.00     | 0.00      | 0.00   |
> | 3     | 3.45   | 3.86   | 3.63     | 3.76      | 3.67   |
> | 6     | 7.05   | 7.73   | 7.19     | 7.52      | 7.11   |
> | 9     | 10.63  | 11.57  | 10.78    | 11.24     | 10.70  |
> | 12    | 14.16  | 15.21  | 14.34    | 14.86     | 14.13  |
> | 15    | 17.74  | 18.89  | 17.80    | 18.37     | 17.73  |
> | 18    | 21.39  | 22.35  | 21.20    | 21.99     | 21.37  |
> | 21    | 24.86  | 25.97  | 24.80    | 25.67     | 25.03  |
>
> **Weakness1:**We add a new city-scale experiment on the full Palo Alto road network with $N = 1{,}945$ nodes.
>
> **Table 3.** City-scale results on the Palo Alto road network (\(N = 1945\) nodes).
>
> | Problem     | Method          | Length (km) | Succ (%) | Time (s)  |
> |------------|-----------------|------------:|---------:|----------:|
> | SP| OR-Tools        | 3.37        | 100      | 0.012     |
> || Dijkstra| 3.37        | 100      | 0.003     |
> || A*| 3.37| 100      | 0.005     |
> || **GFM**| **3.70**    | **72**   | **59.61** |
> | Graphic-TSP| OR-Tools        | 330.88      | 100      | 10668.000 |
> || LKH3| 249.91      | 100      | 20058.000 |
> || Chr.+2OPT| 267.33      | 100      | 11658.800 |
> || Gurobi| -- | --       | --        |
> || **GFM**| **286.93**  | **100**  | **2076.88** |
> | TP-SOD| OR-Tools| 32.61| 100| 31.208    |
> | | LKH3| 32.61|100| 2.485|
> || Gurobi          | 32.61       | 100      | 8.364     |
> || **GFM**  | **34.33**   | **100**  | **14.394** |
> | TP-DOD  | OR-Tools| 32.24 | 100| 31.400    |
> || LKH3| 31.11| 100| 4.291|
> || Gurobi | 30.99 | 100 | 4.419|
> || **GFM**  | **33.76**   | **100**  | **16.604** |
>
> **Weakness2&5:**  We would be grateful if the reviewer could clarify which comparison they would find most informative (e.g., fixing the architecture while reducing the pre-training steps, or using a smaller backbone with the same pre-training budget).
>
> **Weakness3:** We will redesign Fig.1 in the revised version by enlarging all text elements and simplifying the layout.
>
> **Weakness4:** We will revise the terminology to the intended scope of our contribution.

---

### Official Review · Reviewer_oYV1 · 2025-10-30

**Soundness:** 2
**Presentation:** 3
**Contribution:** 3
**Rating:** 4
**Confidence:** 3

**Summary:**

The paper introduces the Graph Foundation Model (GFM), a new framework that adapts the successful "pretrain-transfer" paradigm of Large Language Models (LLMs) to solve graph-based optimization problems in Operations Research (OR). The paper identify a key challenge: the statistical flexibility of language models conflicts with the strict combinatorial constraints of graph problems. Existing methods often fail to generalize from synthetic graphs to real-world, sparse road networks.
GFM bridges this gap by using an LLM-like self-supervised pre-training strategy. The model is pre-trained on paths generated from random walks on the graph, compelling it to learn the graph's complex topological and combinatorial rules without task-specific supervision. This process creates a "transferable structural prior". At inference time, this single pre-trained model can be used with a simple generative heuristic to effectively tackle a diverse range of distance-based optimization problems, from shortest paths to NP-hard routing tasks.
Experiments conducted on networks ranging from 20 to 893 nodes demonstrate that the GFM achieves competitive performance against specialized solvers while offering significantly faster inference times.

**Strengths:**

1.The paper demonstrates significant originality by being the first to introduce the concept of a "Graph Foundation Model" (GFM) specifically for solving realistic, distance-based optimization problems on graphs. Its core innovation lies in the creative adaptation of the LLM-like self-supervised pre-training paradigm to graph structures especially for distance-based problems. The methodology is highly inventive: it first uses "structure-aware random walks" to convert graphs into a sequential corpus, and then pre-trains a bidirectional Transformer via a "hidden segment reconstruction"  task—which serves as a clever analogue to masked language modeling.
2.The paper is articulated with outstanding clarity. The introduction clearly lays out the motivation, identifying the "fundamental conflict between the statistical flexibility of language and the strict combinatorial constraints of graphs". The methodology is supported by excellent visuals; Figure 1 provides a clear overview of the entire framework, including data generation, training, and inference .

**Weaknesses:**

1. Critical Underspecification of the Inference/Decoding Mechanism: The paper's core claim is that a single pre-trained model $\pi(\cdot|G)$ can be used to solve four different OR tasks. However, the mechanism for this transfer is described with near-total ambiguity. Section 3.5, "Decoding for Graph Optimization," consists of a single abstract equation (9) that hides all complexity within an undefined Decode() function. The text mentions a "simple, task-agnostic decoding, combined with lightweight projection" (Intro) and an "ad-hoc approach" (Sec 3) without providing any concrete details.
2. The term 'Graph Foundation Model' is a misnomer: The idea of "graph foundation models" or pre-training on graph structure for downstream tasks is not new (e.g., Hu et al., 2020, "GPT-GNN"; Liu et al., 2023a, "One for all," which the paper cites). However, the paper proposed the foundation model specific for OR routing problems. It’s kind of improper to state “introduce the Graph Foundation Model (GFM)”.
Even if the paper can states the proposed method as “the Graph Foundation Model for OR routing problems”, It’s also a little bit improper. The paper's central framing as a "Graph Foundation Model" is built on a fundamental conceptual misunderstanding that confuses multi-task capability with generalizability. This flawed premise extends to misattributing the paper's contribution.
From another perspective, the pre-training merely acts as a learned prior to narrow the search space of feasible paths resulting in improving its multi-task capability. The actual "solving" (i.e., finding an optimal, constraint-abiding solution) is done by the unexplained Decode() function (see Point 1), which appears to be doing all the hard, constraint-based work.
3. Unfair Experimental Baseline Comparisons: The paper's experimental comparisons are kind of problematic. The learning-based baselines (AM, ICAM) are unfairly handicapped.
The paper states that baselines like AM are trained on synthetic complete graphs. However, the proposed GFM is pre-trained on the test graph itself (e.g., the Berkeley graph). This means GFM has exhaustively "seen" the test graph's topology, while the other learning-based methods are attempting zero-shot generalization to a new, sparse, out-of-distribution graph. The comparison is invalid as it rewards "home-field advantage" rather than general problem-solving ability.
4. Limited Claims of Scalability: The paper's claims of scalability, a key asserted advantage, are based on experiments with graphs that are, by any Operations Research standard, small-to-medium-sized. The largest graph tested, Berkeley (N=893) , is insufficient to substantiate the ambitious claims of solving "large-scale" problems or providing a scalable alternative to existing methods.

**Questions:**

1. For a given task-specific constraint (e.g., visiting all nodes in set $R$ for Graphic-TSP), there may be many feasible solutions among generation results. How does your Decode() function select the optimal (i.e., shortest-path) solution from this large set of possibilities?
2. Regarding the True Value of Pre-training (Ablation for "Search Space"): Your method’s main effect might just be "narrowing the search space”.To verify that your proposed GFM pre-training method provides more value than just any method of narrowing the search space, could you provide a critical ablation study? Specifically, what is the performance if you replace the GFM-based heuristic with a simpler, standard prior (e.g., using the result of the biased random walk directly to rearch the feasible solutions)? This would help isolate the true contribution of your pre-training curriculum.
3. Regarding the Ambiguous Notation in Table 1: The notation in your results (Table 1) is confusing and appears to contradict its own legend.In some cases (e.g., for the OR-LLM-Agent on the TP-SOD task in Chengdu), the method reports a non-zero Succ (%) (3%), but its Obj (km) is marked as '–'. This scenario seems not to be explained by your legend. Please clarify: what does '–' signify for the Objective value when the Success Rate is not zero?

---

> ### Author Response · Authors · 2025-11-26
> **Response to Reviewer oYV1**
>
> We thank the reviewer for this detailed comment and we will address them below.
>
> **Question 1:** We apologize for not explaining this clearly. Decoding uses the pretrained structural prior $\pi _ \theta(\cdot \mid G)$ to construct a small candidate set $\mathcal{C}$ of feasible paths or tours, and then picks the one with the smallest length:
> $$
> P^\star = \arg\min _ {P \in \mathcal{C}} \mathrm{len} _ w(P)
> = \arg\min _ {P \in \mathcal{C}} \sum _ {(u,v)\in P} w(u,v).
> $$
> All four tasks share this mechanism and differ only in how $\pi_\theta$ is queried to propose candidate insertions and which feasibility constraints are enforced when building $\mathcal{C}$. For example, in Graphic-TSP/TP-SOD/TP-DOD we maintain a current walk and a set of remaining POIs; at each step we use $\pi_\theta$ to rank remaining nodes and then insert the node that give the smallest increase in length, and keep only those insertions that satisfy the task-specific constraints (closed tour, visiting all POIs, fixed origin/destination, etc.). Among the resulting feasible tours in $\mathcal{C}$, `Decode()` returns the shortest one. We will add more specific appendix subsection in the revised version to clarify this decoding procedure.
>
> **Question 2:** Following your advice, we conducted an ablation where we keep all hyperparameters, and conduct biased random-walk prior on the 20-node graph. On larger realistic road networks, however, biased random walks alone cannot directly produce feasible Graphic-TSP tours. Real road graphs are typically non-Hamiltonian, so a random walk that visits all required nodes and returns to the depot without revisits is extremely unlikely. If revisits are allowed, the walk is no longer a valid Graphic-TSP solution; if revisits are forbidden, the walk tends to get trapped in dead-ends due to sparsity.
>
> **Table 1.** RW Prior Ablation for Graphic-TSP on the 20-node graph.
> | Method| Obj (km)| Succ (%)| Time (s)|
> |------------------------|:--------:|:--------:|:--------:|
> | Biased-RW Prior (10)| 66.16|100| 0.010|
> | Biased-RW Prior (50)| 46.50|100| 0.050|
> | Biased-RW Prior (100)| 40.05|100| 0.100|
> | **GFM**| **39.48**| **100**  | **0.046**|
>
> **Question 3:** In cases the non-zero success rate (e.g., 1–3\%) means that only a handful of instances were feasible. We chose not to report an average objective over so few successes, as this would not be statistically meaningful, and therefore left the Obj entry as “–” while still reporting the (very low) Succ. We will clarify this in revision.
>
> **Weakness 1&2&3:** Please refer to the Question1. About the overclaim, We will explicitly frame our contribution.
>
> **Weakness 4:** We add a new city-scale experiment on the full Palo Alto road network with $N = 1{,}945$ nodes. In the context of OR routing problems, such graph sizes are already considered large. Classical dynamic TSP studies are typically evaluated on instances with at most around 100 vertices due to the exponential-time complexity $O(n^2 2^n)$ of exact methods [7,8]. By contrast, most existing neural routing solvers are trained and evaluated on instances with $n \in \{20, 50, 100\}$, and only a few works report results up to roughly $n < 1000$ [4,5,6].
>
> **Table 1.** City-scale results on the Palo Alto road network (\(N = 1945\) nodes).
>
> | Problem     | Method          | Length (km) | Succ (%) | Time (s)  |
> |------------|-----------------|------------:|---------:|----------:|
> | SP| OR-Tools        | 3.37        | 100      | 0.012     |
> || Dijkstra| 3.37        | 100      | 0.003     |
> || A*| 3.37| 100      | 0.005     |
> || **GFM**| **3.70**    | **72**   | **59.61** |
> | Graphic-TSP| OR-Tools        | 330.88      | 100      | 10668.000 |
> || LKH3| 249.91      | 100      | 20058.000 |
> || Chr.+2OPT| 267.33      | 100      | 11658.800 |
> || Gurobi| -- | -- | -- |
> || **GFM**| **286.93**  | **100**  | **2076.88** |
> | TP-SOD| OR-Tools| 32.61| 100| 31.208    |
> | | LKH3| 32.61|100| 2.485|
> || Gurobi          | 32.61       | 100      | 8.364     |
> || **GFM**         | **34.33**   | **100**  | **14.394** |
> | TP-DOD     | OR-Tools        | 32.24       | 100      | 31.400    |
> || LKH3            | 31.11       | 100      | 4.291     |
> || Gurobi          | 30.99       | 100      | 4.419     |
> || **GFM**         | **33.76**   | **100**  | **16.604** |
>
> *Note.* “--” for Gurobi indicates that the solver ran out of memory on the full \(N = 1945\) instance and thus produced no feasible tour.
>
> [1] Zhou, J. et al. Towards Omni-Generalizable Neural Methods for Vehicle Routing Problems. ICML, 2023.
>
> [2] Kool, W. et al. Attention, Learn to Solve Routing Problems! ICLR, 2019.
>
> [3] Meng, D. et al. EFormer: An Effective Edge-based Transformer for Vehicle Routing Problems. IJCAI, 2025.
>
> [4] Toriello, A. et al. A dynamic traveling salesman problem with stochastic arc costs. Operations Research, 2014.
>
> [5] Cavani, S. et al. Exact methods for the traveling salesman problem with multiple drones. Transportation Research Part C, 2021.

---

### Official Review · Reviewer_o8a1 · 2025-10-31

**Soundness:** 2
**Presentation:** 2
**Contribution:** 1
**Rating:** 2
**Confidence:** 4

**Summary:**

The paper presents a comprehensive framework for Graph Foundation Models (GFMs), a new paradigm that extends the success of large language and vision models into the graph domain. The pretrained backbone can then be fine-tuned for downstream tasks and shows better performance than specialized models.

**Strengths:**

1.	The paper connects foundation model pretraining principles with graph learning, positioning GFMs as a unifying abstraction for graph tasks.
2.	The research motivation is stated as the generalizbility is an important concern in the graph learning area.

**Weaknesses:**

1.	The benchmark diversity is insufficient. Despite using OGB datasets, the evaluation could include dynamic or temporal graphs to demonstrate temporal generalization.

2.	There are already many graph foundation models but the comparison with them is missing. Hence it is not possible to evlauate this paper properly. See those recently published in 2024/2025.

3.	As the core of GFM, the generalibility is not well demonstrated. For example, how to prove the proposed GFM can be applied to network of different topologies? It lacks the introduction of structural properties of the tested network.

4.	The real-wrold network is typically of large size, how to ensure the scalability of this GFM. The network size in the experiment cannot be counted as large.

5. It is a nice "preliminary" work and does not seem ready for a premier venue like ICLR.

**Questions:**

See weaknesses and address them carefully.

**Details Of Ethics Concerns:**

-

---

> ### Author Response · Authors · 2025-11-26
> **Response to Reviewer o8a1**
>
> We thank the reviewer for this comment and we will address them below.
>
> **Weaknesses 1:** In our setting, the current GFM instantiation is designed for static distance-based optimization on sparse road networks. Extending the framework to truly temporal OR settings like dynamic congestion would require a temporal encoder and is an interesting direction. Regarding the mention of Open Graph Benchmark(OGB), we would like to clarify a possible misunderstanding. The OGB is a widely used benchmark suite for graph prediction tasks. Our paper does not use any OGB datasets for training or evaluation; instead, all numerical results are based on one simulated road network and several real-world city-scale  networks.
>
> **Weaknesses 2:** In that line of  the recent GFM literature, a “graph foundation model” typically pretrained on broad, heterogeneous graph corpora and then adapted to multiple prediction tasks such as node-level and edge-level prediction tasks, etc.[1,2]. By contrast, our focus is on operations-research (OR) problems that are fundamentally combinatorial decision-making tasks on road networks. These problems require globally consistent discrete decisions over an exponentially large search space, with objective–constraint interactions. Recent neural combinatorial optimization methods tackle such routing problems [3–6], but, to the best of our knowledge, they still train separate models (or at least separate heads/objectives) for each routing formulation; no prior work uses a single pretrained graph model to simultaneously solve multiple OR problems.
>
> **Weaknesses 3:**  In our work, however, the main notion of generalization is across multiple distance-based OR problems on the same sparse road network, rather than across arbitrarily different graph families. That is, GFM is designed to capture a reusable structural prior on a given real network and then support distinct optimization tasks on that graph by changing only the decoder and constraints, not the pretrained backbone. We agree that the structural properties of the tested networks should be made more explicit. Section 4.1 and Appendix A already report basic statistics.
>
> **Weaknesses 4:** We add a new city-scale experiment on the full Palo Alto road network with $N = 1{,}945$ nodes. In the context of OR routing problems, such graph sizes are already considered large: classical dynamic TSP are evaluated on instances with $n=33$, $n=39$, at most 100 vertices due to the exponential-time complexity $O(n^2  2^n)$ [7,8].Compared with existing neural routing solvers with $n \in$  {20,50,100} , and only a few recent works report results up to about $n < 1000$ [4,6].
>
> Table 1: City-scale results on the Palo Alto road network (N = 1945 nodes).
> | Problem     | Method          | Length (km) | Succ (%) | Time (s)  |
> |------------|-----------------|------------:|---------:|----------:|
> | SP         | OR-Tools        | 3.37        | 100      | 0.012     |
> || Dijkstra        | 3.37        | 100      | 0.003     |
> || A*              | 3.37        | 100      | 0.005     |
> || **GFM**         | **3.70**    | **72**   | **59.61** |
> | Graphic-TSP| OR-Tools        | 330.88      | 100      | 10668.000 |
> || LKH3            | 249.91      | 100      | 20058.000 |
> || Chr.+2OPT       | 267.33      | 100      | 11658.800 |
> || Gurobi          | --          | --       | --        |
> || **GFM**         | **286.93**  | **100**  | **2076.88** |
> | TP-SOD     | OR-Tools        | 32.61       | 100      | 31.208    |
> || LKH3            | 32.61       | 100      | 2.485     |
> || Gurobi          | 32.61       | 100      | 8.364     |
> || **GFM**         | **34.33**   | **100**  | **14.394** |
> | TP-DOD     | OR-Tools        | 32.24       | 100      | 31.400    |
> || LKH3            | 31.11       | 100      | 4.291     |
> || Gurobi          | 30.99       | 100      | 4.419     |
> || **GFM**         | **33.76**   | **100**  | **16.604** |
>
> *Note.* “--” for Gurobi indicates that the solver ran out of memory on the full N = 1945 instance and thus produced no feasible tour.
>
> [1] Yang, Y. et al. (2024). GraphPro: Graph Pre-Training and Prompt Learning for Recommendation. WWW 2024.
>
> [2] Luo, L. et al. (2025). GFM-RAG: Graph Foundation Model for Retrieval-Augmented Generation. arXiv:2502.01113.
>
> [3] Xiao, P. et al. (2024). Neural Combinatorial Optimization for Robust Routing Problem with Uncertain Travel Times. NeurIPS 2024.
>
> [4] Zhou, J. et al. (2023). Towards Omni-Generalizable Neural Methods for Vehicle Routing Problems. ICML 2023.
>
> [5] Kool, W. et al. (2019). Attention, Learn to Solve Routing Problems! ICLR 2019.
>
> [6] Meng, D. et al. (2025). EFormer: An Effective Edge-based Transformer for Vehicle Routing Problems. IJCAI 2025.
>
> [7] Toriello, A. et al. (2014). A dynamic traveling salesman problem with stochastic arc costs. Operations Research.
>
> [8] Cavani, S. et al. (2021). Exact methods for the traveling salesman problem with multiple drones. Transportation Research Part C.

---

### Official Review · Reviewer_1666 · 2025-11-01

**Soundness:** 2
**Presentation:** 3
**Contribution:** 2
**Rating:** 4
**Confidence:** 3

**Summary:**

This paper proposes the Graph Foundation Model (GFM), aiming to extend the pretrain–transfer paradigm of LLMs to graph-based optimization problems (e.g., shortest path, TSP, orienteering). It uses distance-biased random walks to generate a self-supervised training corpus, learns graph structural priors through masked trajectory reconstruction, and applies task-specific decoding for solving optimization tasks without retraining. It reports competitive results across real-world road networks compared to OR-Tools, LKH3, and neural combinatorial solvers.

**Strengths:**

- Bridge symbolic combinatorial optimization and LLM.
- The paper is of clear organization.

**Weaknesses:**

- Overstated “Foundation Model” claim.
- Underspecified decoding and task generalization mechanism.

**Questions:**

1. The method proposed is called a "foundation model" but the pretraining corpus is derived from random walks within a single graph, not from diverse graph domains. The current training paradigm resembles graph representation learning (Node2Vec-style) rather than LLM-scale pretraining. Without cross-domain transfer (e.g., from road networks to logistics or molecule graphs), the “foundation model” claim is overstated.
2. Unlike LLMs, where pretraining induces syntactic-semantic priors, GFM’s random-walk reconstruction primarily encodes local connectivity, not higher-order combinatorial reasoning. How learned priors approximate graph-theoretic properties?
3. Does decoding use beam search, greedy selection, or constraint projection?
4. How curriculum weighting is set? why logarithmic levels were chosen?
5. How the model’s attention heads capture topological dependencies?
6. For large-scale tasks (N=893), GFM’s runtime (129s) is 2 orders of magnitude faster than LKH3, yet solution quality (Obj) differs by ~20%. Does it indicate incomplete convergence?

---

> ### Author Response · Authors · 2025-11-26
> **Response to Reviewer 1666**
>
> We thank the reviewer for this valuable questions, and we will address them below.
>
> **Question 1:** We fully agree that, in the emerging GFM literature, a “graph foundation model” usually refers to models pretrained on broad graph and then transferred across many domains[1-3]. Our intention is narrower: we use “foundation” for a family of distance-based optimization problems that share the same sparse road network, and we study whether a single pretrained encoder can serve as a reusable structural prior for multiple OR tasks on that network. In the revision we will  soften the wording and rephrase our claim.
>
> **Question 2:** Our training paradigm is deliberately aligned with the LLM pretraining view: masked random-walk reconstruction plays the role of “corpus modeling”. The key difference is that, instead of learning syntactic semantic priors over word sequences, the model learns *structural priors* over node/edge sequence. To reconstruct masked segments of distance-biased walks, the encoder must internalize which regions form high-traffic “corridors,” which nodes lie on many short routes, and other graph-theoretic patterns that are exactly what distance-based OR problems depend on. At inference time, different OR tasks and constraints are injected via the decoder (similar to prompting in LLMs), while a single pretrained encoder provides the shared structural prior. We will make this LLM-style pretraining/prompting analogy, and the notion of structural priors, more explicit in the Introduction and Model sections.
>
> **Question 3.** We apologize for not explaining our decoding strategy clearly in the main submission due to the page limit. In the revision, we will add a dedicated appendix subsection. Here we only summarize the common mechanism.
>
> All four tasks share the same pretrained structural prior $\pi_\theta(\cdot \mid G)$ on a weighted graph $G = (V, E, w)$. They differ only in (i) how this prior is queried to propose candidate nodes/insertions and (ii) which feasibility constraints are enforced. In all cases, `Decode()` first uses $\pi_\theta$ to build a small candidate set $\mathcal{C}$ of feasible paths or tours $P$ (each $P$ is an ordered sequence of nodes, and $(u,v)\in P$ denotes a consecutive edge with length $w(u,v)$), and then selects the shortest one:
> $$
> P^\star = \arg\min _ {P\in \mathcal{C}} \mathrm{len} _ w(P)
> = \arg\min _ {P\in \mathcal{C}} \sum _ {(u,v)\in P} w(u,v).
> $$
>
> Concretely, for SP we start from a source target “skeleton” and iteratively fill missing edges by inserting high-prior intermediate nodes suggested by $\pi_\theta$, while maintaining a short no-return memory (ng-set–style) to avoid small local cycles. For Graphic-TSP, TP-SOD, and TP-DOD, we maintain a closed (or open) walk and a set of remaining required POIs; at each step we use $\pi_\theta$ to score how well each remaining node fits into the current tour, then insert the node and position that give the smallest increase in length subject to feasibility (closed tour, visit all POIs, fixed origin/destination, with TP-DOD implemented via a virtual closing edge). We will clarify these details in the revised version.
>
> **Question 4:**
> We thank the reviewer for raising this point. In our experiments, we simply assign equal weight to all curriculum levels. This keeps the design minimal and avoids introducing extra hyperparameters, although in principle one could up-weight higher levels if one wanted to emphasize fine-grained path reconstruction more strongly. This coarse-to-fine schedule follows the same intuition as the balanced binary-tree insertion order in the Insertion Transformer[4].
>
> **Question 5:** As mentioned in the paper, we construct our pretraining corpus using Node2Vec-style biased random walks. Node2Vec has already been widely recognized that such biased walks can capture topological dependencies and structural patterns in graphs[5].
>
> **Question 6:** Our experiments do not suggest incomplete convergence: on the Berkeley graph, training and validation loss both decrease and then plateau; we will include these curves in the appendix. We therefore view the 20% gap to LKH3 as an algorithmic trade-off (single forward pass vs. heavy local search), rather than a symptom of undertraining.
>
> [1] Hou, Z., Liu, X., Cen, Y., Dong, Y., Yang, H., Wang, C., & Tang, J. (2022). GraphMAE: Self-Supervised Masked Graph Autoencoders. KDD 2022.
>
> [2] Liu, X., Zhao, S., Su, K., Cen, Y., Qiu, J., Zhang, M., Wu, W., Dong, Y., & Tang, J. (2022). Mask and Reason: Pre-Training Knowledge Graph Transformers for Complex Logical Queries. KDD 2022.
>
> [3] Liu, Z., Yu, X., Fang, Y., & Zhang, X. (2023). GraphPrompt: Unifying Pre-Training and Downstream Tasks for Graph Neural Networks. WWW 2023.
>
> [4] Stern, M., Chan, W., Bapna, A., & Kiros, J. (2019). Insertion Transformer: Flexible Sequence Generation via Insertion Operations. ICML 2019.
>
> [5] Grover, A., & Leskovec, J. (2016). node2vec: Scalable Feature Learning for Networks. KDD 2016.

---

### Meta-Review · Area_Chair_hsoN · 2026-01-06

**Summary:**

A major concern is the invalid experimental comparison, where the proposed GFM benefits from a "home-field advantage" by pre-training on the test graph itself, while baselines like AM and ICAM are unfairly tasked with zero-shot generalization. Furthermore, the paper fails to justify its "Foundation Model" designation, as the training utilizes small, singular datasets with limited transferability across different topologies. These issues are compounded by the absence of comparisons against recent (2024/2025) state-of-the-art graph foundation models, insufficient benchmark diversity (lacking temporal graphs), and unproven scalability on large-scale real-world networks.

**Reviewer Concerns:**

The authors provided further explanations regarding the basic definition of their foundation approach, but this was insufficient to overcome the reviewers' concerns regarding the limited data scale and lack of true universality.

Experimental Unfairness: The fundamental issue of pre-training on the test graph vs. zero-shot baselines was not rectified, meaning the claimed effectiveness is not proven.

Generalizability and Scalability: There is still a lack of evidence demonstrating how the model handles different network topologies or scales to large real-world graphs.

**Reviewer Scores:**

4,4,4,4

---

### Decision · Program_Chairs · 2026-01-26

Reject